# The Synergistic Effect of Carbon Black/Carbon Nanotube Hybrid Fillers on the Physical and Mechanical Properties of EPDM Composites after Exposure to High-Pressure Hydrogen Gas

**DOI:** 10.3390/polym16081065

**Published:** 2024-04-11

**Authors:** Hyunmin Kang, Jongwoo Bae, Jinhyok Lee, Yumi Yun, Sangkoo Jeon, Nakkwan Chung, Jaekap Jung, Unbong Baek, Jihun Lee, Yewon Kim, Myungchan Choi

**Affiliations:** 1Elastic Material Research Group, Korea Institute of Materials Convergence Technology, Busan 47154, Republic of Korea; khmin0402@gmail.com (H.K.); jwbae@kimco.re.kr (J.B.); jhlee@kimco.re.kr (J.L.); ymyun@kimco.re.kr (Y.Y.); 2Hydrogen Energy Group, Korea Research Institute of Standards and Science, Daejeon 34113, Republic of Korea; sangku39@kriss.re.kr (S.J.); nk.chung@kriss.re.kr (N.C.); jkjung@kriss.re.kr (J.J.); ubbaek@kriss.re.kr (U.B.); ljh93@kriss.re.kr (J.L.); kyw9687@kriss.re.kr (Y.K.)

**Keywords:** EPDM, carbon black, multi-wall carbon nanotube, hybrid, physical and mechanical properties, resistance to high-pressure hydrogen gas

## Abstract

This study investigated the synergistic effect of carbon black/multi-wall carbon nanotube (CB/MWCNT) hybrid fillers on the physical and mechanical properties of Ethylene propylene diene rubber (EPDM) composites after exposure to high-pressure hydrogen gas. The EPDM/CB/CNT hybrid composites were prepared by using the EPDM/MWCNT master batch (MB) with 10 phr CNTs to enhance the dispersion of CNTs in hybrid composites. The investigation included a detailed analysis of cure characteristics, crosslink density, Payne effect, mechanical properties, and hydrogen permeation properties. After exposure to 96.3 MPa hydrogen gas, the hydrogen uptake and the change in volume and mechanical properties of the composites were assessed. We found that as the MWCNT volume fraction in fillers increased, the crosslink density, filler–filler interaction, and modulus of hybrid composites increased. The hydrogen uptake and the solubility of the composites decreased with an increasing MWCNT volume fraction in fillers. Moreover, after exposure to hydrogen gas, the change in volume and mechanical properties exhibited a diminishing trend with a higher MWCNT volume fraction. We conclude that the hybridization of CB and CNTs formed strong filler–filler networks in hybrid composites, consequently reinforcing the EPDM composites and enhancing the barrier properties of hydrogen gas.

## 1. Introduction

Rubber is currently one of the most extensively used products in industry and society, with uses in areas such as transportation, the automotive industry, and infrastructure. This is due to the unique properties of rubber, which is both elastic and viscous. Furthermore, typical rubber compound formulations consist of several ingredients that are added to improve these physical properties, affect vulcanization, and improve processability [1,2].

Carbon black (CB) is the most commonly used filler for reinforcement in the rubber industry. CB leads to improvements in the tensile strength, modulus, hardness, and abrasion resistance of rubber vulcanizates due to its high specific surface activity, which promotes extensive rubber–filler interactions [3]. However, when CB is added above a certain content, the mechanical properties, such as tensile strength and elongation at break, decrease because CB particles agglomerate and have poor dispersion [4]. Additionally, the processability decreases due to the increase in the viscosity of the composites. Therefore, it is necessary to find alternative fillers to replace the high CB contents.

To improve the mechanical properties of rubber composites with high CB contents, a hybrid system is necessary. The hybrid system combines two or more different types of fillers in rubber composites. In particular, the combination of nanofillers and conventional CB filler has received attention in research aimed at developing rubber materials with enhanced properties. Hybrid composites with nanofillers could retain the superior properties of all fillers, and those also produced synergistic effects in rubbers due to the fillers’ compatibility and cooperative interactions at the nanoscale level [3,5,6,7]. Accordingly, hybrid composites with the inclusion of nanosized particles have attracted the attention of many researchers. Senthivel, K et al. [8] and Ismail, H et al. [9] reported that the cure time, rubber–filler interaction, and mechanical properties of NBR increased with increasing halloysite nanotubes (HNTs) content due to the combined effect of CB/HNTs. Furthermore, in other studies of a hybrid filler system of carbon black and organoclay, the mechanical properties and thermal stability improved compared to a pure polymer [10,11].

In the field of hybrid systems based on CB and nanoparticles, carbon nanotubes (CNTs) are regarded as representative nanoparticles that are compatible with CB. CNTs exhibit high mechanical strength, electrical, thermal, and barrier properties, attributed to their high aspect ratio, and they thus offer efficient reinforcement for composites [12,13,14]. Furthermore, CNTs can achieve comparable properties with significantly lower carbon nanotube contents compared to high carbon black contents [15]. Previously, researchers investigated the mechanical [6,16,17,18,19,20,21], thermal [22,23,24], and electrical [25,26,27,28] properties of various rubbers to explore the synergies between CB and CNT. They observed that the hardness, modulus, thermal stability, and electrical conductivity increased compared when utilizing CB alone. This was attributed to the synergistic effects of the combination of CB and CNT. However, despite the various contributions in the literature on the mechanical and physical properties of composites with CB/CNT hybrid systems, few studies have investigated the gas barrier properties of composites with CB/CNT hybrids. Furthermore, most research on gas barrier properties has focused on single filler systems, with limited attention on hybrid systems. S. Wen et al. [29] and Y.Q. Gill et al. [30] investigated the effect of the topological structures of nanofillers on the gas barrier properties of polymers reinforced by three types of carbon-based nanofillers: zero-dimensional CB, one-dimensional CNT, and two-dimensional graphene. With the addition of nanofillers, the gas barrier properties of polymers were all improved to a certain extent. However, different trends in the enhancement of gas barrier properties were seen in the composites reinforced by different topologies of nanofillers. Among these, two-dimensional graphene exhibits the highest gas barrier properties, followed by the one-dimensional and then the zero-dimensional. This is attributed to the high aspect ratio and formation of a more complete filler network, particularly in two-dimensional graphene at a low filler content. Despite its advantageous properties, the industrial use of graphene remains limited due to its high cost. Conversely, one-dimensional CNTs, with a high aspect ratio, also improve the gas barrier properties of rubber composites and are industrially advantageous. Therefore, further research is necessary to investigate the gas barrier properties of composites with hybrid systems utilizing CNTs.

Typically, materials with excellent gas barrier properties are required for seals in the nozzle section of hydrogen tanks for fuel cell vehicles (FCVs) and hydrogen fuel stations (HFSs). Rubber materials have low hydrogen permeation properties, as well as excellent thermal and chemical resistance, resulting in their wide use as sealing materials [31,32,33]. Among them, EPDM in particular is the most widely used material for seals such as O-rings and gaskets due to its excellent heat and chemical resistance, flexibility over a wide temperature range, and good hydrogen gas barrier properties. In previous articles dealing with rubber composites with conventional fillers such as silica and CB [34,35,36], high filler loadings in the composites contribute to enhanced hydrogen barrier properties owing to the increased reinforcement and crosslink density. Furthermore, as the filler loading increased, the filler volume fraction increased and the mobility of the rubber chain decreased, resulting in excellent gas barrier properties.

In this study, CB and multi-wall carbon nanotube (MWCNT) were employed as reinforcing fillers in a hybrid system of EPDM composites. To observe the effects of MWCNTs in the hybrid filler system of composites, the volume fraction of total fillers in rubber for all composites was kept constant while varying the MWCNTs content. Initially, we investigated to analyze the effects of CB/MWCNT hybrid fillers on the crosslink density, mechanical, and hydrogen permeation properties of the hybrid composites without exposure to hydrogen gas. Furthermore, assessments were conducted on the volume change, hydrogen uptake, and the change in mechanical properties of EPDM hybrid composites after exposure to 96.3 MPa hydrogen gas.

## 2. Materials and Methods

### 2.1. Materials

Ethylene propylene diene rubber (EPDM, KEP 2480, ethylene content: 57.5%, ethylidene norbornene content: 8.9%, Kumho Polychem Co., Ltd., Seoul, Republic of Korea) was used as the rubber matrix. Carbon black (N330, Orion Engineered Carbons Co., Ltd., Senningerberg, Luxembourg) and an EPDM/multi-wall carbon nanotube masterbatch (EPDM/MWCNT MB, MWCNT: K-Nanos BEP 120, MWCNT content: 10 phr, Kumho Petrochemical Co., Ltd., Seoul, Republic of Korea) were used as reinforcing fillers. The curing agent and accelerator we utilized were dicumyl peroxide (DCP, Perkadox BC-FF, Nouryon, Amsterdam, The Netherlands) and triallyl cyanurate (TAC, FARIDA TACE, Hunan Farida Technology Co., Ltd., Changsha, China). Zinc oxide and stearic acid were supplied by PJ CHEMTEK Ltd. (Yangsan, Republic of Korea) and LG Household & Health Care (Seoul, Republic of Korea), respectively.

### 2.2. Preparation of EPDM Hybrid Composites

EPDM/CB/CNT hybrid composites with different MWCNT contents containing 0, 1, 3, and 5 parts per hundred of rubber (phr) were prepared using a mechanical mixing method. The detailed formulation is shown in Table 1. The volume of the total amount of fillers in rubber for all composites is 15.8 vol%. To demonstrate the effects of MWCNTs in the hybrid filler system of the rubber composites, the volume fractions of MWCNT in total filler are varied between 0, 4.3, 12.9, and 21.5%. To control the volume fraction of MWCNT in fillers and keep constant EPDM content in the composites, EPDM/MWCNT MB and EPDM were used. A peroxide curing system was employed for the preparation of the EPDM/CB/CNT composites.

The composites were prepared through a two-step mixing process. In the first mixing step, EPDM rubber was mixed with CB and EPDM/MWCNT MB in an internal mixer (Measuring Mixer 30, Brabender GmbH & CO. KG, Duisburg, Germany) with a starting temperature of 80 °C and a mixing dump temperature of 130 °C. In the second mixing step, the curing agent and accelerator were added to the EPDM/CB/CNT hybrid composites by using an 8-inch two open roll mill (PK-RM20140930, Pungkwang CO., Hwaseong, Republic of Korea).

Then, the composites were vulcanized via compression molding in a hot press set at a temperature of 160 °C. The compression molding time was determined by the t_90_ value using Rubber Process Analyzer (RPA, RPA elite, TA Instruments, New Castle, DE, USA).

The hybrid composites are denoted by MWCNT-χ, where χ represents the corresponding MWCNT volume fraction in fillers present in the sample. For example, MWCNT-12 is an EPDM composite with a 12.9% MWCNT volume fraction in fillers.

### 2.3. Viscosity and Curing Behavior

The Mooney viscosity of composites was measured in a Mooney Viscometer (DMV-200C, Dae Kyung Engineering Co., Ltd., Jeju, Republic of Korea). The measurement was taken at 125 °C, requiring 1 min heating time and 4 min measuring time.

The curing characteristics of the composites were analyzed using RPA following the ASTM D 2084 standard at 160 °C. The oscillation frequency and amplitude were 1.67 Hz and 1°, respectively.

### 2.4. Crosslink Density

The crosslink density of the composites was determined using a volume swelling test. We placed 15 × 15 × 2 mm^3^ rectangle samples in tetrahydrofuran (THF) for 72 h at room temperature, and then they were removed. The swollen samples were immediately weighed after being cleaned of any remaining adhering THF from the surfaces of the samples. The weights of the samples were determined by using an electronic balance with an accuracy of 0.001 g.

The crosslink density was calculated using the Flory–Rehner equation as follows [37,38]:(1)$$V=\frac{1}{2M_{C}}=-\frac{\ln\left(1-V_{1}\right)+V_{1}+\chi V_{1}^{2}\;}{2\rho_{r}V_{0}(V_{1}^{\frac{1}{3}}-\frac{V_{1}}{2})}$$
(2)$$V_{1}=\frac{\frac{W_{d}-W_{f}}{\rho_{r}}}{\frac{W_{d}-W_{f}}{\rho_{r}}+\frac{W_{s}-W_{d}}{\rho_{s}}}$$
where *V* is the crosslink density (mol/g), *M_C_* is the average molecular weight between crosslinking points (g/mol), *V*_0_ is the molar volume of the solvent (cm^3^/mol), *V*_1_ is the volume fraction of rubber in the swollen gel at equilibrium, *W_d_* is the weight of the unswollen sample, *W_f_* is the weight of the filler in the sample, *W_s_* is the weight of the swollen sample, *ρ_r_* is the density of the rubber, *ρ_s_* is the density of the solvent, and *χ* is the polymer–solvent interaction parameter (*χ* = 0.501) [39].

### 2.5. Payne Effect

The degree of filler–filler interaction was evaluated by measuring the dynamic storage modulus (G′) of the EPDM hybrid composites using RPA. The test was carried out at 100 °C with a frequency of 1.67 Hz and variable strain from 0.1% to 100%. The delta G′ at low and high strain is used to represent the degree of filler–filler interaction.

### 2.6. Transmission Electron Microscopy (TEM)

The hybrid network structure between CB and MWCNT was examined by a Cs corrected scanning transmission electron microscope (Cs-STEM, NEO ARM, JEOL Ltd., Tokyo, Japan) operated at an acceleration voltage of 200 kV. The ultrathin sections of samples were prepared by a Cryo-Ultramicrotome (PTPC&CRX, RMC, Tucson, AZ, USA) fitted with a diamond knife and cooled at −60 °C by liquid nitrogen and put onto copper grids.

### 2.7. Mechanical Properties

The hardness of the composites was assessed using a Shore A durometer (ASKER Type A, KOBUNSHI KEIKI Co., Ltd., Kyoto, Japan), following the ASTM D 2240. Meanwhile, the mechanical properties of the composites were determined on a universal testing machine (UTM, Instron 3345, Instron Ltd., Norwood, MA, USA). The tensile properties were assessed using test method A of ASTM D 412. The test specimens were of the Die C type of ASTM D 412, and the crosshead speed was set to 500 mm/min. A minimum of 5 samples were utilized for measuring the mechanical properties, and the average result was reported with its standard deviation.

### 2.8. Hydrogen Permeation Properties

The hydrogen permeation properties were assessed utilizing a differential pressure method based on Fick’s diffusion law and Henry’s gas solubility law. Samples of the circular shape with a 45 mm diameter were cut from the sheet. Testing was carried out using 99.9999% pure hydrogen gas at room temperature. The samples were positioned between the upper and bottom cells, and a consistent gas-permeable area was maintained by using the porous support. The pressures at which the upper and bottom cells were pumped out were 0.1 kPa and 0.001 kPa, respectively. A 100 kPa pressure was reached by pumping hydrogen gas into the upper cell, and the pressure in the bottom cell was recorded and plotted until it reached 1.3 kPa. The hydrogen permeability coefficient, diffusivity coefficient, and solubility coefficient of samples were calculated based on the results obtained [40].

### 2.9. Remaining Hydrogen Content

By utilizing 99.999% hydrogen gas in a high-pressure cylindrical chamber for 24 h at room temperature, the maximum hydrogen exposure of 96.3 MPa was attained. To remove extra gases, a three times hydrogen purge at 5 MPa was carried out after the samples were placed in the chamber. After the purge, hydrogen gas was pumped into the chamber at a rate of approximately 5 MPa/min, up to a pressure of 96.3 MPa. For the gas sorption of a sample with a thickness of about 2.5 mm, hydrogen gas exposure for 24 h is enough to reach the equilibrium state. At a rate of approximately 1 MPa/s, the decompression to atmospheric pressure was carried out [41].

To determine the hydrogen uptake, we used a volumetric analysis technique using a graduated cylinder described elsewhere [42]. Hydrogen absorbed inside the polymer under high-pressure conditions was released to the outside when decompressed to atmospheric pressure. The elapsed time was recorded from the instant (t = 0) at which the high-pressure hydrogen chamber reached atmospheric pressure. After decompression, the samples and the filler particles on the mesh removed from the chamber were loaded into the empty space of the graduated cylinder partially immersed in distilled water, and then the graduated cylinder was sealed with a plug. The hydrogen released from the sample pushed the water to the bottom, and this reduced water level was detected. By reading this change in water level inside a graduated cylinder versus the time elapsed after depressurization, the amount of hydrogen released from the sample can be measured. Because the samples are removed from the chamber and then loaded into the graduated cylinder, a time lag between decompression and the start of the measurement exists. Thus, the hydrogen uptake was determined by compensating for the amount of hydrogen released during the time lag through the offset value of a diffusion analysis program [42].

### 2.10. Volume Change and Mechanical Properties of the EPDM Hybrid Composites after Exposure to High-Pressure Hydrogen Gas

The volume change of composites was measured both before and after exposure to hydrogen gas at a pressure of 96.3 MPa for 24 h at room temperature. After decompression, the volume change values of the EPDM/CB/MWCNT hybrid composites at 1 h and 72 h were obtained as follows:(3)$$\Delta V\;\left(\%\right)=\frac{V_{f}-V_{i}}{V_{i}}\times 100$$
where *V_f_* is the volume of the sample at 1 h or 72 h after decompression, and *V_i_* is the initial volume of the sample without exposure to hydrogen gas.

A tensile test was carried out utilizing standard test specimens of the Die C type specified in ASTM D 412 to investigate the effect of exposure to high-pressure hydrogen gas on the mechanical properties of composites. In order to prepare the test specimens of composites exposed to hydrogen gas, sheets with dimensions of 100 mm × 240 mm × 2 mm were placed into the chamber, which was filled with 96.3 MPa hydrogen gas for 24 h. Test specimens were prepared by cutting them from the swollen sheets after decompression. The change in the mechanical properties (Δ*M*) of EPDM/CB/MWCNT composites at 1 h or 72 h was calculated as follows:(4)$$\Delta M\;\left(\%\right)=\frac{M_{a}-M_{i}}{M_{i}}\times 100$$
where *M_a_* is the mechanical properties at 1 h or 72 h after decompression, and *M_i_* is the initial mechanical properties of unexposed hydrogen gas.

## 3. Results and Discussion

### 3.1. Viscosity and Curing Behavior

The effects of the MWCNT volume fraction in fillers on the Mooney viscosity and cure characteristics of the EPDM/CB/MWCNT composites are shown in Table 2. This shows that the Mooney viscosity of the composites was slightly increased with an increasing MWCNT volume fraction in fillers. This resulted from the flow restriction of the highly entangled CNTs. Because CNTs have a high aspect ratio and there is relatively poor interfacial adhesion between CNTs and CB, it is difficult to disentangle and disperse CNTs in rubber, especially at a high CNT volume fraction [24].

Figure 1 shows the vulcanization characteristics of EPDM/CB/MWCNT hybrid composites. The rheometric parameters including minimum torque (M_L_), maximum torque (M_H_), delta torque (ΔT = M_H_ − M_L_), scorch time (t_10_), optimum cure time (t_90_), and cure rate index are shown in Table 2. This reveals that M_L_ and M_H_ of the hybrid composites increased with an increasing MWCNT volume fraction in fillers. This can be attributed to the physical interaction between the MWCNTs and CB, which constrained the mobility of the rubber molecular chain [12]. Furthermore, ΔT of the composites filled with CB/CNTs was higher than that for the composites without CNTs, but it was independent of CNT loading, likely due to poor dispersion of the MWCNTs in rubber. With an increasing MWCNT volume fraction in fillers, t_10_ and t_90_ of the hybrid composites decreased and the cure rate index of those increased. This indicated that MWCNTs could accelerate the curing reaction of the rubber matrix in the composites.

### 3.2. Crosslink Density of EPDM/CB/MWCNT Composites

A swelling experiment was necessary for determining the crosslinking density of composites. The results of our swelling experiment of EPDM hybrid composites for different MWCNT volume fraction in fillers are shown in Figure 2. This demonstrates that the swelling ratios of EPDM/CB/MWCNT hybrid composites gradually decreased with an increasing MWCNT volume fraction in fillers. The swelling ratio of the composite without CNTs was 121.9%, and those of composites with MWCNT volume fraction in fillers from 4.3 to 21.5% were 117.6%, 109.4%, and 104.9%, respectively.

The crosslink density of EPDM/CB/MWCNT hybrid composites was then estimated using the Flory–Rehner equation based on the swelling experiments. We found that the crosslink density of EPDM/CB/MWCNT hybrid composites increased with an increasing MWCNT volume fraction in fillers. Generally, the outcome of the Flory–Rehner equation comprises both physical and chemical crosslinking in the composites. Because a consistent amount of chemical crosslinking was used when preparing the composites, it was assumed that their amount of chemical crosslinking was similar regardless of the MWCNT volume fraction in fillers. Therefore, we surmised that the increase in the crosslink density of the composites with increasing MWCNT volume fraction in fillers was significantly affected by the physical crosslinking resulting from interfacial adhesion between CNTs and CB.

### 3.3. Payne Effect of EPDM Hybrid Composites

The Payne effect is considered to be a specific characteristic of the stress–strain behavior of rubber composites with fillers, particularly carbon black. It is characterized by the relationship between the amplitude of the applied strain and the storage and loss moduli. The storage modulus (G′) rapidly decreases with increasing amplitude over a certain critical strain amplitude, saturating at significant deformations. Conversely, the loss modulus reaches its maximum within the range where the storage modulus decreases [43]. In research, the Payne effect is quantified by the specific value ΔG′, which is the difference between the maximum value and the minimum value of G′ [44]. The Payne effect is dependent upon the filler content of the material and disappears with the unfilled elastomers. Figure 3a shows the strain dependency of the G′ of the CB/MWCNT-filled EPDM. The G′ at low strain increases with an increasing MWCNT volume fraction in fillers. Garcia, D. B. et al. [45] showed that the G′ at low strain in the CNT-filled composite is higher than in the CB-filled composite, despite the CNT content being much lower than that of CB. This can be attributed to the fact that CNTs have a higher available surface to interact with the rubber chains compared to CB, which has an amorphous shape. Consequently, as the MWCNT volume fraction in fillers increased, a strong filler network between the MWCNTs and CB formed in the composites [46]. Figure 3b shows that the Payne effect (ΔG′) of composites increases with the addition of MWCNTs in EPDM/CB/MWCNT composites. ΔG′ of the composite without CNTs is 349 kPa, while those of composites with MWCNT volume fractions in fillers from 4.3 to 21.5% are 418, 517, and 639 kPa, respectively. Furthermore, ΔG′ of the hybrid composite with an MWCNT volume fraction in fillers of 21.5% increases by about twice that of the composite without MWCNTs. This can be explained by the creation of denser and stronger filler–filler interactions as the MWCNT volume fraction in fillers increases [15].

The structures of EPDM/CB/MWCNT hybrid composites were observed by TEM analyzing the CB and MWCNTs filler network behavior. TEM microphotographs of EPDM/CB/MWCNT hybrid composites with varying MWCNT content are shown in Figure 4. It can be observed that as the MWCNT volume fraction in fillers increased, many MWCNTs were present between the CB aggregates, leading to enhanced connectivity with the CB aggregates. The MWCNTs acted as bridges, linking the CB aggregates to create a hybrid filler network by both fillers. Therefore, with the increase in MWCNT volume fraction in fillers, higher filler–filler interactions and strong filler networks were evident.

### 3.4. Physical and Mechanical Properties of EPDM/CB/MWCNT Composites

The stress–strain curves of EPDM/CB/MWCNT hybrid composites with different MWCNT volume fractions in fillers are shown in Figure 5, and the tensile properties of the hybrid composites are summarized in Table 3. We can surmise from these that the addition of CNTs enhanced the hardness and 100% modulus of the composites and reduced the tensile strength and elongation at break. The hardness of the composites gradually increased with an increasing MWCNT volume fraction in fillers. Meanwhile, the addition of CNTs caused a slight reduction in the tensile strength of composites, regardless of the MWCNT volume fraction in fillers, which was observed to be dependent on the CNT volume fraction. Additionally, the elongation at break decreased from 340% to 271% as the MWCNT volume fraction in fillers increased. It can be attributed to the increased crosslink density of the composites as the MWCNT volume fraction in fillers increased, resulting in increased stiffness.

The 100% modulus of EPDM/CB/MWCNT hybrid composites increased with an increasing MWCNT volume fraction in fillers. For instance, the 100% modulus of MWCNT-21 was 6.8 MPa, a value that was significantly increased (about 79% more) compared to the composites without MWCNTs. This observation indicates that the presence of MWCNTs increased the stiffness of the rubber macromolecular chain more than the effect of CB. It is well-known that nanoparticle fillers with a higher aspect ratio than CB can provide a physical crosslink in rubber [12], and these results show that the CB/CNT hybrid system exhibits a strong reinforcing effect on the EPDM composites.

### 3.5. Hydrogen Permeation Properties of EPDM/CB/MWCNT Hybrid Composites

The permeation properties of EPDM/CB/MWCNT hybrid composites for hydrogen gas are shown in Figure 6 and Table 4. These demonstrate that the MWCNT volume fraction in fillers did not affect the permeability and diffusivity coefficients of the composites. In general, those are influenced by the free volume of the rubber in the composites [47,48] and, in this study, the free volume was consistent because the total volumetric number of fillers in the composites was kept constant. Meanwhile, the solubility coefficient of the composites decreased with an increasing MWCNT volume fraction in fillers, as shown in Figure 6c.

To observe the difference in hydrogen uptake between CB and CNTs, pure CB and CNTs were individually placed in tea bags and exposed to hydrogen gas at 10 MPa for 48 h. The results are shown in Figure 6d, displaying that the $w_{H_{2}}/w_{sample}$ which demonstrates the weight ratio of hydrogen uptake and sample for CNTs was 218.2 μg/g, whereas that of CB was higher at 326.5 μg/g. These results indicated that hydrogen uptake is dependent on the molecular structure and shape of fillers.

### 3.6. Hydrogen Uptake of EPDM/CB/MWCNT Composites

The hydrogen uptake of EPDM/CB/MWCNT composites is presented in Figure 7. Figure 7a shows the hydrogen emission concentration curve of EPDM/CB/MWCNT hybrid composites at different MWCNT volume fraction in fillers. As the MWCNT volume fraction in fillers increases, the equilibrium value and the time to reach the hydrogen emission concentration of composites decrease. Moreover, after 10,000 s, the hydrogen emission concentration of composites reaches equilibrium, regardless of the MWCNT volume fraction in fillers. Figure 7b shows the hydrogen uptake of EPDM/CB/MWCNT hybrid composites, which was calculated by using the hydrogen emission concentration. The hydrogen uptake of composites decreases with an increasing MWCNT volume fraction in fillers. For instance, compared to the composite without MWCNTs, the hydrogen uptake decreases by 8.8% and 30.0% at MWCNT-4 and MWCNT-21, respectively. These trends are similar to the results of the solubility coefficients of the composites, a finding we had expected because, as explained previously, the hydrogen uptake is influenced by the solubility coefficients of fillers.

### 3.7. Change in Mechanical Properties of EPDM/CB/MWCNT Hybrid Composites after 96.3 MPa Hydrogen Gas Exposure

The volume changes of the EPDM/CB/MWCNT hybrid composites at 1 h and 72 h after decompression are shown in Figure 8a. At 1 h after hydrogen decompression, the volume change of composites decreased with the increasing MWCNT volume fraction in fillers. Notably, the volume change of MWCNT-21 was 4%. Then, as the MWCNT volume fraction in fillers continued to increase, the penetration of hydrogen molecules into the composites was observed to be more difficult.

We found that 72 h after hydrogen decompression was the time at which the equilibrium value of the hydrogen emission concentration was reached. At this time, the volume change of the composites was less than 1%, which recovered to the value before the exposure to high-pressure hydrogen gas. From these results, we can infer that the pressure difference in the composites led the absorbed hydrogen molecules to be released, which is why the volume change of the composites decreased with a longer decompression period.

The changes in the mechanical properties of the EPDM/CB/MWCNT composites are shown in Figure 8. At 1 h after decompression, the mechanical properties of composites decreased compared to those before exposure to 96.3 MPa hydrogen gas. The degradation of the mechanical properties could be explained by rapid hydrogen decompression, which can promote the formation of cavities inside rubber [49]. We found that the change in the mechanical properties of the composites decreased with an increasing MWCNT volume fraction in fillers. This we attribute to the increasing ratio leading to a reduction in hydrogen uptake in the composites, producing a corresponding decrease in volume change. We investigated the correlation between the change in the mechanical properties of the composites at 1 h after decompression and the hydrogen uptake, as shown in Figure 9. The changes in the mechanical properties and hydrogen uptake were found to take an inverse linear correlation, indicating that a decrease in hydrogen uptake resulted in a decrease in the change in mechanical properties. Moreover, the change in the 100% modulus after exposure to hydrogen gas was found to be significantly affected by hydrogen uptake. This can be attributed to the considerable deviation in the tensile strength and elongation at break of rubber materials, while the 100% modulus exhibits less deviation.

At 72 h after decompression, the mechanical properties of the composites with MWCNTs were recovered to similar values as those before the exposure to high-pressure hydrogen gas, which suggests the recovery of the volume change at a sufficient time after decompression. Meanwhile, the mechanical properties of the composites without MWCNTs were decreased, which was due to microbubbles and cracks within the rubber composites [50].

## 4. Conclusions

In this study, we prepared EPDM composites filled with a combination of carbon black/multi-wall carbon nanotube (CB/MWCNT) hybrid fillers. During our investigations, we observed an increase in the crosslink density and the Payne effect of composites with higher MWCNT volume fraction in fillers, which we attribute to the strong network between MWCNT and CB. Furthermore, the hardness and 100% modulus significantly improved with the addition of MWCNTs.

Meanwhile, the solubility coefficients for hydrogen gas in the composites decreased with an increasing MWCNT volume fraction in fillers because MWCNT has lower hydrogen uptake than CB. Accordingly, as the MWCNT volume fraction in fillers increased, there was a notable decrease in hydrogen uptake, volume change, and a change in the mechanical properties of the composites after exposure to 96.3 MPa hydrogen gas.

Consequently, we conclude that EPDM/CB/MWCNT hybrid composites, particularly those with a high MWCNT volume fraction in fillers, are appropriate for applications such as sealing material used in hydrogen fuel cell vehicles and stations, mainly due to the formation of the strong filler network between MWCNT and CB, and resistance to high-pressure hydrogen gas.

## Figures and Tables

**Figure 1 polymers-16-01065-f001:**
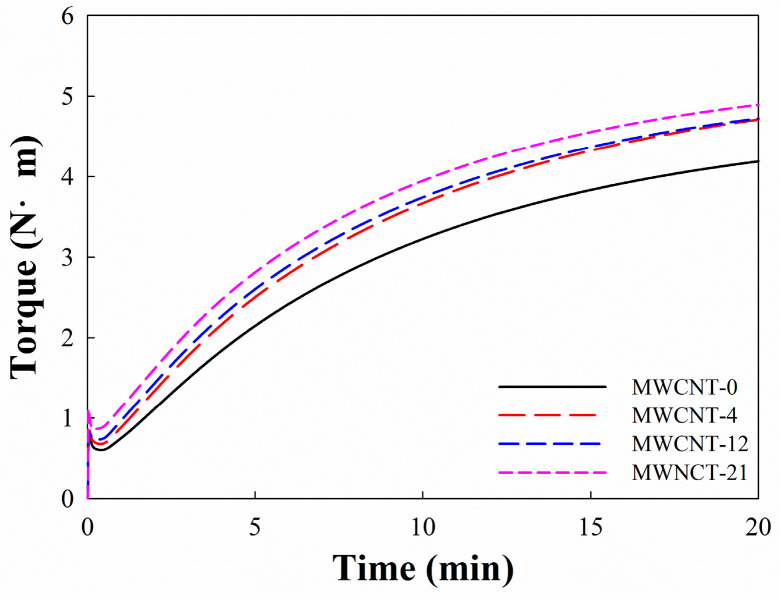
Curing curves of EPDM/CB/MWCNT hybrid composites.

**Figure 2 polymers-16-01065-f002:**
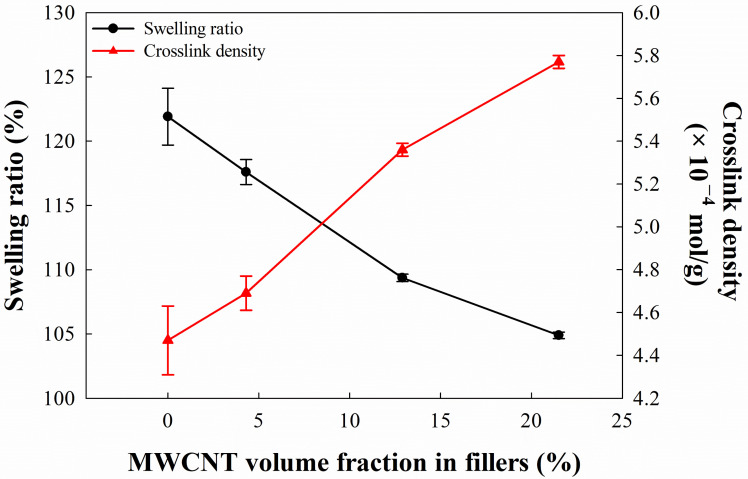
Swelling ratio and crosslink density of EPDM/CB/MWCNT hybrid composites.

**Figure 3 polymers-16-01065-f003:**
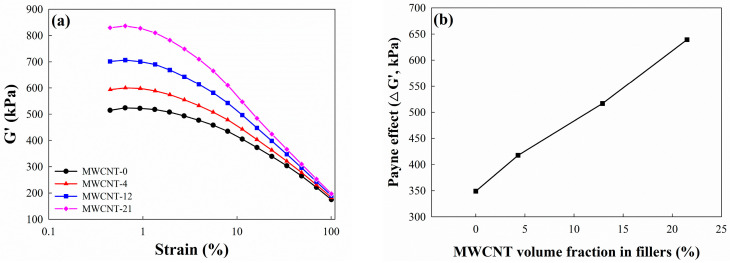
Payne effect of EPDM/CB/CNT hybrid composites: (**a**) storage modulus as a function of strain at 100 °C; (**b**) ΔG′ of EPDM/CB/MWCNT hybrid composites according to MWCNT volume fraction in fillers.

**Figure 4 polymers-16-01065-f004:**
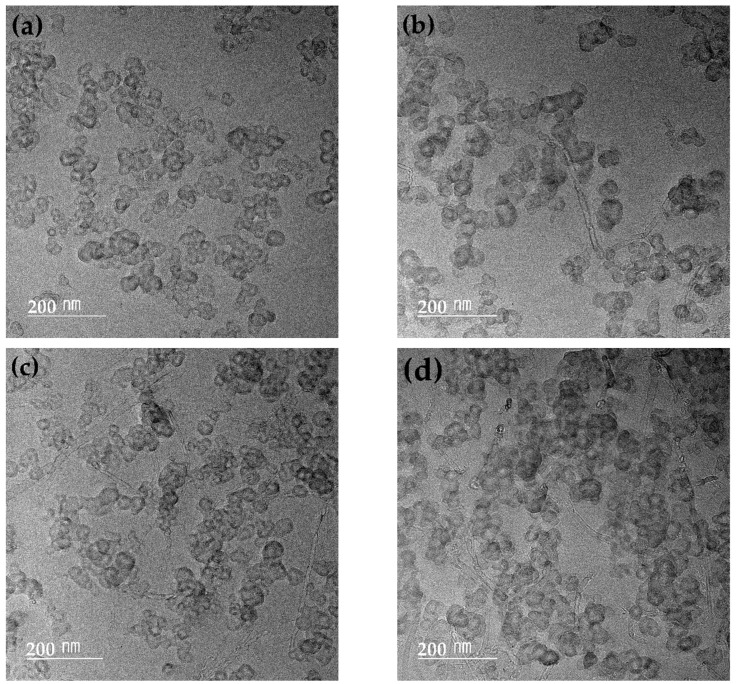
TEM micrographs of EPDM/CB/MWCNT hybrid composites with different MWCNT volume fraction in fillers: (**a**) MWCNT-0; (**b**) MWCNT-4; (**c**) MWCNT-12; and (**d**) MWCNT-21.

**Figure 5 polymers-16-01065-f005:**
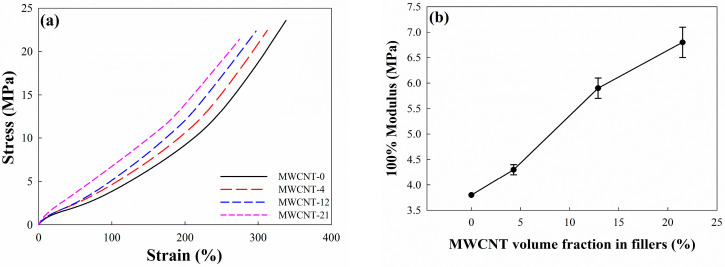
Mechanical properties of EPDM hybrid composites: (**a**) strain–stress curve of EPDM/CB/MWCNT composites; (**b**) 100% modulus of CB/MWCNT-filled EPDM composites.

**Figure 6 polymers-16-01065-f006:**
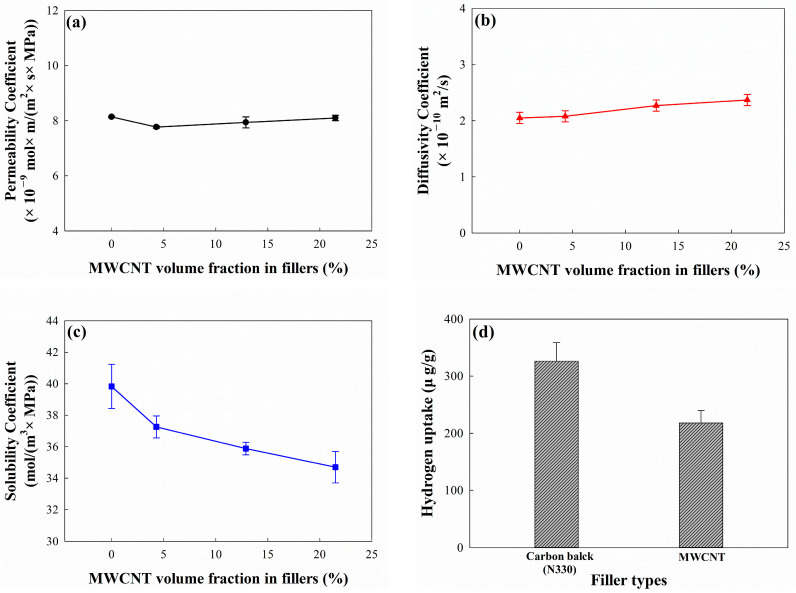
Hydrogen permeability of EPDM/CB/MWCNT hybrid composites according to MWCNT volume fraction in fillers: (**a**) permeability coefficient; (**b**) diffusivity coefficient; (**c**) solubility coefficient; and (**d**) hydrogen uptake of CB and MWCNT.

**Figure 7 polymers-16-01065-f007:**
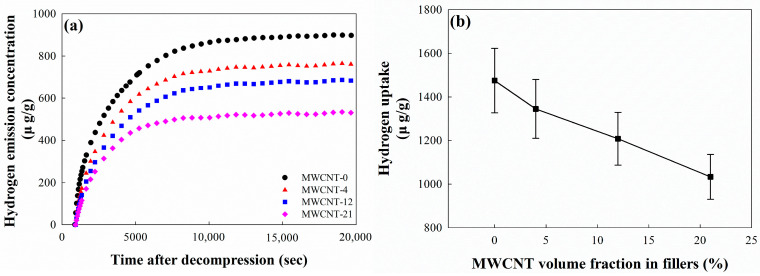
Hydrogen uptake of CB/MWCNT-filled EPDM hybrid composites: (**a**) emission concentration after exposure to high-pressure hydrogen gas; and (**b**) hydrogen uptake of EPDM/CB/MWCNT composites.

**Figure 8 polymers-16-01065-f008:**
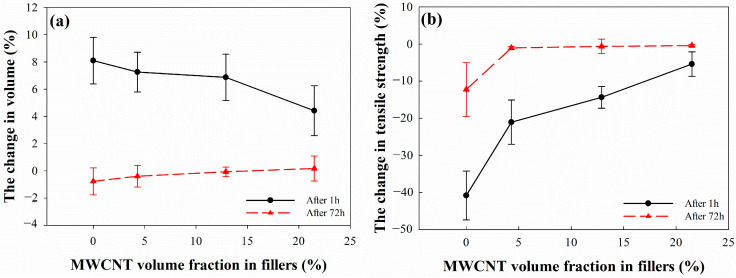
Effect of MWCNT volume fraction in fillers on the mechanical property changes in EPDM composites exposed to 96.3 MPa hydrogen gas for 24 h: (**a**) change in volume; (**b**) change in tensile strength; (**c**) change in elongation at break; and (**d**) change in 100% modulus.

**Figure 9 polymers-16-01065-f009:**
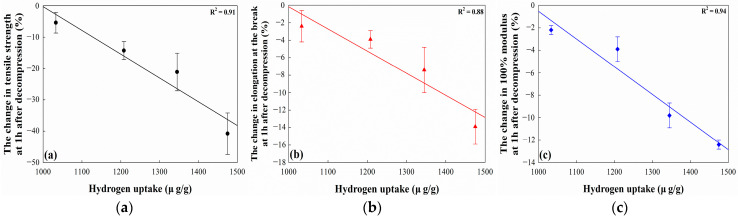
Correlation between the change in the mechanical properties at 1 h after decompression and hydrogen uptake: (**a**) the change in tensile strength; (**b**) the change in elongation at break; and (**c**) the change in 100% modulus.

**Table 1 polymers-16-01065-t001:** Formulations of EPDM/CB/MWCNT composites according to the MWCNT volume fractions in fillers.

Ingredients (phr)	MWCNT-0	MWCNT-4	MWCNT-12	MWCNT-21
EPDM	100	90	70	50
EPDM/MWCNT MB	0	11 ^(a)^	33 ^(b)^	55 ^(c)^
Carbon black	40	38.3	34.8	31.4
Zinc oxide	3	3	3	3
Stearic acid	1	1	1	1
DCP	1.5	1.5	1.5	1.5
TAC	1.0	1.0	1.0	1.0
Volume fraction of MWCNT in fillers (%)	0	4.3	12.9	21.5

^(a)^ EPDM 10 phr + MWCNT 1 phr; ^(b)^ EPDM 30 phr + MWCNT 3 phr; ^(c)^ EPDM 50 phr + MWCNT 5 phr.

**Table 2 polymers-16-01065-t002:** Rheometric parameters of the EPDM hybrid composites.

Properties	Parameters	MWCNT-0	MWCNT-4	MWCNT-12	MWCNT-21
Physical property	Mooney viscosity (MU)	90.3	102.6	106.9	113.0
Cure characteristics	M_L_ (N·m)	0.60	0.68	0.73	0.86
	M_H_ (N·m)	4.19	4.71	4.72	4.89
	Δt (N·m)	3.59	4.03	3.99	4.03
	T_10_ (min)	1.58	1.43	1.34	1.27
	T_90_ (min)	14.95	14.81	14.54	14.30
	Cure rate index (N·m/min)	0.30	0.34	0.36	0.37

**Table 3 polymers-16-01065-t003:** Physical and mechanical properties of CB/MWCNT-filled EPDM hybrid composites.

Properties	MWCNT-0	MWCNT-4	MWCNT-12	MWCNT-21
Hardness (Shore A)	68	70	74	75
Tensile strength (MPa)	23.4 ± 1.0	21.6 ± 0.4	21.1 ± 1.9	21.3 ± 1.5
Elongation at break (%)	340 ± 6.9	319 ± 3.1	282 ± 21.8	271 ± 19.1
100% modulus (MPa)	3.8 ± 0.0	4.3 ± 0.1	5.9 ± 0.2	6.8 ± 0.3

**Table 4 polymers-16-01065-t004:** Summary of the hydrogen gas permeation properties of the composites.

MWCNT Volume Fraction in Fillers (%)	Permeability Coefficient (×10^−10^ mol/m·s·MPa)	Diffusivity Coefficient (×10^−9^ m^2^/s)	Solubility Coefficient (mol/m^3^·MPa)
0	8.2 ± 0.05	2.1 ± 0.1	39.8 ± 1.4
4.3	7.8 ± 0.04	2.1 ± 0.1	37.3 ± 0.7
12.9	7.9 ± 0.2	2.3 ± 0.1	35.9 ± 0.4
21.5	8.1 ± 0.1	2.4 ± 0.1	34.7 ± 1.0

## Data Availability

Data are contained within the article.

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
