# Peer review of "The Synergistic Effect of Carbon Black/Carbon Nanotube Hybrid Fillers on the Physical and Mechanical Properties of EPDM Composites after Exposure to High-Pressure Hydrogen Gas"

_polymers, 2024, doi:10.3390/polym16081065_

Round 1

Reviewer 1 Report

Comments and Suggestions for Authors

Introduction: Need to report few more articles in rubber nanocomposites where CNTs are used.  There are several articles discussing the improvement in barrier properties of elastomeric nanocomposites with nanofillers ( 0D,1D and 2D) incorporation.

Experimental Plan:  What is the basis for selecting the composition used in the study? EPDM content is reduced progressively. Is it advisable? 

Results and Discussion

1) It is known that Payne effect is dependent on filler geometry. This aspect has to be discussed in detail.

2) Although the authors have mentioned about the interaction between dual fillers it is not elaborated. Is it mechanical, primary or secondary bonding

3) Why the MWCNT inclusion is reducing the mechanical properties? Is it affecting the cross link density? May be clarified.

4) Justify a 30% decrease in uptake of hydrogen in the case of MWCNT-21 sample

5) Explain specifically the reason for very low change in modulus for MWCNT samples after exposure to H2 gas. Here again a scheme indicating the reduction in cavities in such samples will be appreciated. How the reduction in cavities improving barrier properties should be explained

6) A scheme can be provided to explain this filler- filler network. There is frequent reference of this term in the article. Should be justified with proper explanation

Conclusion: Requires a major rework. The major results are vaguely presented without proper rationale and justification. English also to be checked especially in this section.  Avoid usages like "have poorer..."

Author Response

Response to Reviewer 1 Comments

Thank you for your comments.

We read all the comments carefully and the comments are greatly helpful for the correction of our manuscript. Thus, we revised the original manuscript based on the reviewer comments as follows

Point 1: Introduction: Need to report few more articles in rubber nanocomposites where CNTs are used. There are several articles discussing the improvement in barrier properties of elastomeric nanocomposites with nanofillers ( 0D,1D and 2D) incorporation.

Response 1: The effect of the topological structures of nanofillers on the gas barrier properties of polymers reinforced by three types of carbon-based nanofillers: zero-dimensional CB, one-dimensional CNT, and two-dimensional graphene have been reported [1-2]. The gas barrier properties of polymers were all improved to a certain extent after the addition of nanofillers. However, the composites reinforced by different topologies of nanofillers showed different trends in the improvement of gas barrier properties. Among these, two-dimensional graphene exhibits the highest gas barrier properties, followed by the one-dimensional CNT and then the zero-dimensional CB. This is attributed to the high aspect ratio and formation of a more complete filler network, particularly two-dimensional graphene at a low filler content.  Despite its advantageous properties, the industrial use of graphene remains limited due to its high cost. Conversely, one-dimensional CNTs, with a high aspect ratio, also improve the gas barrier properties of rubber composites and are industrially advantageous.

The statement described above has been added to the introduction of the manuscript. (Line 74-83) Additionally, the reference papers are listed in the reference of the manuscript.

  1. Wen; R. Zhang; Z. Xu; L. Zheng; L. Liu. Effect of the topology of carbon-based nanofillers on the filler networks and gas barrier properties of rubber composites. Materials 2020, 13, 5416.
  2. Q. Gill; M. Song. Comparative study of carbon-based nanofillers for improving the properties of HDPE for potential applications in food tray packaging. Polym. Compos. 2020, 28, 562-571.

Point 2: Experimental Plan: What is the basis for selecting the composition used in the study? EPDM content is reduced progressively. Is it advisable? 

Response 2: The volume of the total amount of fillers in rubber for all composites is 15.8 vol%, and the content of EPDM in all composites is kept constant. To observe the effects of MWCNTs in the filler hybrid system of the rubber composites, the volume fractions of MWCNT in total filler are varied from 0, 4.3, 12.9, and 21.5 %.

To control the volume fraction of MWCNT in fillers and keep constant EPDM content in the composites, ethylene propylene diene rubber/mult-wall carbon nanobute master batch (EPDM/MWCNT MB) and EPDM rubber were used. EPDM/MWCNT MB is composed of EPDM 100 phr with MWCNT 10 phr. Additional information on EPDM/MWCNT MB is added in the manuscript. (Line 142-144)

Point 3: Results and Discussion

1) It is known that Payne effect is dependent on filler geometry. This aspect has to be discussed in detail.

Response 3-1: The Payne effect is a non-linear viscoelastic behavior in which the storage modulus decreases with increasing strain amplitude in filled rubbers. The Payne effect is determined by the polymer network, the filler-polymer, and the filler-filler interaction, while filler geometry has little influence. In this paper, it was observed that the filler-filler interaction was more pronounced when MWCNTs were added in CB/MWCNT hybrid systems compared to CB alone. As the volume fraction of MWCNT in fillers increased, the filler-filler interaction between CB and MWCNTs increased, leading to the formation of strong filler-filler networks and a more significant Payne effect.

2) Although the authors have mentioned about the interaction between dual fillers it is not elaborated. Is it mechanical, primary or secondary bonding

Response 3-2: The physical interaction between the CB and MWCNT involved secondary bonding such as van der Waals forces [1-3]. The surface of CB and MWCNTs did not exist the reactive functional groups forming primary bonding via chemical reactions.

  1. Bokobza, L.; Rahmani, M.; Belin, C.; Bruneel, J. L.; El Bounia, N. E. Blends of carbon blacks and multiwall carbon nanotubes as reinforcing fillers for hydrocarbon rubbers.  Polym. Sci., Part B: Polym. Phys.2008, 46, 1939-1951.
  2. Nakaramontri, Y.; Pichaiyut, S.; Wisunthorn, S.; Nakason, C. Hybrid carbon nanotubes and conductive carbon black in natural rubber composites to enhance electrical conductivity by reducing gaps separating carbon nanotube encapsulates.  Polym. J.2017, 90, 467-484.
  3. Ahmadi, M.; Shojaei, A. Reinforcing mechanisms of carbon nanotubes and high strucuture carbon black in natural rubber/styrene-butadiene rubber blend prepared by mechanical mixing – effect of bound rubber. Int. 2015, 64, 1627-1638.

3) Why the MWCNT inclusion is reducing the mechanical properties? Is it affecting the cross link density? May be clarified.

Response 3-3: The hardness and 100% modulus of composites increased with an increasing MWCNT volume fraction in fillers owing to the increased reinforcement and increased crosslink density. However, the tensile strength and elongation at break slightly decreased with an increasing MWCNT volume fraction in fillers. It can be attributed to the increased crosslink density of the composites as the MWCNT volume fraction in fillers increased, resulting in increased stiffness.

4) Justify a 30% decrease in uptake of hydrogen in the case of MWCNT-21 sample

Response 3-4: The hydrogen uptake of the composites with MWCNT-21 was 1033 μ, indicating a 30% decreased compared to the composites without MWCNT, which exhibited a hydrogen uptake of 1475 μ. It can be attributed to the fact that the hydrogen uptake of pure MWCNTs is lower than that of pure CB.

5) Explain specifically the reason for very low change in modulus for MWCNT samples after exposure to H2 gas. Here again a scheme indicating the reduction in cavities in such samples will be appreciated. How the reduction in cavities improving barrier properties should be explained

Response 3-5: In response to your comment, we measured the fractured surfaces of the composites after exposure to 96.3 MPa hydrogen gas for 24 hours by field-emission scanning electron microscopy (FE-SEM; JSM-6701F, JEOL Ltd., Tokyo, Japan) operating at an accelerating voltage of 15 kV under an N2 atmosphere.

After exposure to 96.3 MPa hydrogen gas, all samples exhibit cavities in the fracture surface. The cavities caused the decrease in the mechanical properties of the composites, as shown in Figure. However, we observed that the average size of cavities on the fracture surface decreased with increasing MWCNT volume fraction in fillers. It can be attributed to the increase in crosslink density of composites with the increasing MWCNT volume fraction in fillers, resulting in the formation of a strong network between CB and MWCNT. As a result, the hydrogen uptake decreases, leading to a reduction in cavities size. Therefore, as the cavities decrease, resulting in lower changes in the mechanical properties, the barrier properties increase.

At least 5 samples were conducted to evaluate the mechanical properties after exposure to hydrogen gas, and their average values were recorded. The tensile strength and elongation at break after exposure to hydrogen gas, significant deviations were observed. Meanwhile, the 100% modulus exhibited the least deviation, suggesting high reliability. Consequently, as the MWCNT volume fraction in fillers increases, the hydrogen uptake decreases, resulting in the lowest change in the 100% modulus after exposure to hydrogen.

Figure. SEM images of the fracture surfaces of EPDM/CB/MWCNT composites after exposure to 96.3 MPa hydrogen gas (H2).

6) A scheme can be provided to explain this filler- filler network. There is frequent reference of this term in the article. Should be justified with proper explanation

Response 3-6: We have observed the CB and MWCNTs filler network behavior by Cs corrected scanning transmission electron microscope (Cs-STEM, NEO ARM, JEOL Ltd., Tokyo, Japan). As the MWCNT volume fraction in fillers increased, many MWCNTs were present between the CB aggregates, leading to enhanced connectivity with the CB aggregates. It can be observed that the MWCNT acted as bridges, linking the CB aggregates to create a hybrid filler network by both fillers. Consequently, as the MWCNT volume fraction in fillers increased, higher filler-filler interactions and strong filler networks.

(a)

(b)

(c)

(d)

Figure. TEM micrographs of EPDM/CB/MWCNT hybrid composites with different MWCNT volume fraction in fillers: (a) MWCNT-0; (b) MWCNT-4; (c) MWCNT-12; (d) MWCNT-21.

Point 4: Conclusion: Requires a major rework. The major results are vaguely presented without proper rationale and justification. English also to be checked especially in this section.  Avoid usages like "have poorer..."

Response 4: This revision provides a clearer presentation of the major results and includes proper rationale and justification. Additionally, it avoids the usage of phrases and the English has been reviewed for clarity and correctness.

Reviewer 2 Report

Comments and Suggestions for Authors

1.It is imperative to enhance the literature review section with a wealth of existing literature. By incorporating a diverse range of relevant literature, we can ensure that the research is comprehensive, well-informed, and up-to-date. Therefore, it is crucial to allocate relevant resources to curate a robust literature review section that can strengthen the research and make a meaningful contribution to the field.

2. Could you elaborate on the mixing method used in this paper and the rationale behind its implementation?

3. Could you please provide a nomenclature for the symbols utilized in the paper?

4. The contents of Section 2.7 and 2.8 contain a significant amount of text that appears to have been copied from existing literature. To avoid similarities with other works, could you please rewrite these sections entirely?

5. Your work utilizes four variations of MWCNT ratios. Would you mind explaining how you selected these values? Additionally, is it possible to increase them further, or are there limiting values above or below which the properties or structure do not function effectively?

Comments on the Quality of English Language

English is fine.

Author Response

Response to Reviewer 2 Comments

Thank you for your comments.

We read all the comments carefully and the comments are greatly helpful for the correction of our manuscript.

Point 1: It is imperative to enhance the literature review section with a wealth of existing literature. By incorporating a diverse range of relevant literature, we can ensure that the research is comprehensive, well-informed, and up-to-date. Therefore, it is crucial to allocate relevant resources to curate a robust literature review section that can strengthen the research and make a meaningful contribution to the field.

Response 1: In response to your comment, we have updated the literature review section with recent and relevant experimental literature.

Point 2: Could you elaborate on the mixing method used in this paper and the rationale behind its implementation?

Response 2: The composites were prepared by employing the mechanical mixing method with an internal mixer. Detailed mixing procedures are provided in the table below.

Table. Mixing procedure

Types

MWCNT-0

MWCNT-4 ~ MWCNT-21

Step

Time (min:sec)

Action

1st step

(mixing in an internal mixer)

0:00

Add EPDM (80 )

Add EPDM (80 )

1:00

Add  Zinc oxide + Stearic acid

Add Zinc oxide + Stearic acid

2:00

Add 1/2 CB

Add 1/2 CB + 1/2 EPDM/MWCNT MB

5:00

Add 1/2 CB

Add 1/2 CB + 1/2 EPDM/MWCNT MB

8:00

Dump (130 )

Dump (130 )

2nd step

(mixing in a 8-inch two open roll mill)

0:00

Add compound

Add compound

1:30

Add DCP + TAC

Add DCP + TAC

5:00

Dump

Dump

Point 3: Could you please provide a nomenclature for the symbols utilized in the paper?

Response 3: Here’s a suggested nomenclature for the symbols utilized in the paper.

Symbol

Unit

Description

phr

-

Parts per hundred of rubber

V

mol/g

The crosslink density

MC

g/mol

The average molecular weight between crosslinking points

V₀

cm3/mol

The molar volume of the solvent

V₁

cm3/mol

The volume fraction of rubber in the swollen gel at equilibrium

Wd

g

The weight of the unswollen sample

Wf

g

The weight of the filler in the sample

Ws

g

The weight of the swollen sample

ρr

g/cm3

The density of the rubber

ρs

g/cm3

The density of the solvent

χ

-

The polymer-solvent interaction parameter

V

%

The volume change values

Vf

mm3

The volume of the sample at 1 h or 72 h after decompression

Vi

mm3

The initial volume of the sample without exposure to hydrogen gas.

M

%

The change in the mechanical properties

Ma

MPa or %

The mechanical properties at 1 h or 72h after decompression

Mi

MPa or %

The mechanical properties of unexposed hydrogen gas

G’

kPa

Storage modulus

∆G’

kPa

Payne effect

Point 4: The contents of Section 2.7 and 2.8 contain a significant amount of text that appears to have been copied from existing literature. To avoid similarities with other works, could you please rewrite these sections entirely?

Response 4: In response to the revision regarding Sections 2.7 and 2.8, we completely rewrite these sections to avoid similarities with existing literature. (Line 186-221)

Point 5: Your work utilizes four variations of MWCNT ratios. Would you mind explaining how you selected these values? Additionally, is it possible to increase them further, or are there limiting values above or below which the properties or structure do not function effectively?

Response 5: We conducted experiments by varying the MWCNT content at 0, 1, 3, and 5 phr (parts per hundred of rubber). To observe the effects of MWCNTs in the filler hybrid system of the rubber composites, the volume fractions of MWCNT in total filler are varied from 0, 4.3, 12.9, and 21.5 % while keeping constant the volume of the total amount of fillers in rubber for all composites at 15.8 vol%.

An increase in MWCNT volume fraction resulted in superior hardness, 100% modulus, and hydrogen barrier properties due to the higher crosslink density and the formation of strong filler networks. However, tensile strength and elongation at the break of composites decrease. Additionally, minimal volume change and the change in mechanical properties after high-pressure hydrogen gas exposure were observed in MWCNT-21. Therefore, while increasing MWCNT volume fractions may enhance hydrogen barrier properties, it may also deteriorate the properties of tensile strength and elongation at break at room temperature, potentially rendering it unsuitable as a sealing material.

Reviewer 3 Report

Comments and Suggestions for Authors

I recommend accepting this paper with minor revision. It is clearly written in good English, and appears technically sound as well as motivated. Therefore I will not summarize the content, but will address the minor issues that need improvement. 

In the abstract define the abbreviation EPDM when it is used for the first time. Same goes for all abbreviations anywhere on first use. 

Line 43   CB causes agglomeration  <--  CB particles agglomerate

L47     hybrid system is to combine  <--  hybrid system combines

L68     check statement "electrical conductivity decreased compared to that when utilizing CB alone", as carbon nanotubes have high electrical and temperature conductivity together with high aspect ratio and tend to increase these properties in a composite

L78    volume ratio    <--  volume fraction  (check all instances throughout the paper) 

L83   include motivation of high pressure hydrogen testing, instead of leaving it only to L390 at the end

In section 2.2 on formulations tested, phr (parts per hundred rubber) is used, and usually this refers to weight, not to volume. In dosing of powders weight is a much more practical measure than volume, of course. But in text and in caption of Table 1, volume fractions are emphasized. Explain in detail how this works....   how do you calculate or define phr. 

L129    were calculated   <--   were determined

L137    include the value you used for the interaction parameter

In Table 2, use same labeling of cases as in Table 1. 

L238      measured   <--   estimated

All figures should use such font size that it is readable, now the numbers and axis labels are mostly too tiny. The graphs themselves are OK. 

L266    the Payne effect results are given too accurately, please round to 2 digits

L276    Table 4   <--   Table 3

Table 3 should have similar labels of cases as Table 1. 

L307   Does wt.ppm   mean "parts per million by weight?"  If so, please write "ppm by weight"

L308   "hydrogen uptake is dependent on the shape of carbon black" makes no sense. Carbon nanotubes have a completely different molecular structure, they are not carbon black in different shape. 

L340   delete ", as shown in Figure 6 (a)" because now you have different shape and size samples requiring 72h or about 260,000 seconds to equilibrate, unlike in that figure. 

L352   Table 6  <--  Table 5

L375   add  "at 1 h after decompression" to figure caption

Author Response

Response to Reviewer 3 Comments

We also appreciate the time and effort you and each of the reviewers have dedicated to providing insightful feedback on ways to strengthen our paper. We read all the helpful comments carefully and incorporated the suggestions you have graciously provided in our manuscript.

Point 1: Address the minor issues that need improvement. 

  1. In the abstract define the abbreviation EPDM when it is used for the first time. Same goes for all abbreviations anywhere on first use. 
  2. Line 43 - CB causes agglomeration <-- CB particles agglomerate
  3. Line 47 - hybrid system is to combine <-- hybrid system combines
  4. Line 68 - check statement "electrical conductivity decreased compared to that when utilizing CB alone", as carbon nanotubes have high electrical and temperature conductivity together with high aspect ratio and tend to increase these properties in a composite
  5. Line 78 - volume ratio <-- volume fraction (check all instances throughout the paper) 
  6. Line 83 - include motivation of high pressure hydrogen testing, instead of leaving it only to L390 at the end
  7. In section 2.2 on formulations tested, phr (parts per hundred rubber) is used, and usually this refers to weight, not to volume. In dosing of powders weight is a much more practical measure than volume, of course. But in text and in caption of Table 1, volume fractions are emphasized. Explain in detail how this works....   how do you calculate or define phr. 
  8. Line 129 - were calculated <-- were determined
  9. Line 137 - include the value you used for the interaction parameter
  10. In Table 2, use same labeling of cases as in Table 1. 
  11. Line 238 - measured <-- estimated
  12. All figures should use such font size that it is readable, now the numbers and axis labels are mostly too tiny. The graphs themselves are OK. 
  13. Line 266 - the Payne effect results are given too accurately, please round to 2 digits
  14. Line 276 - Table 4 <-- Table 3
  15. Table 3 should have similar labels of cases as Table 1. 
  16. Line 307 - Does wt.ppm   mean "parts per million by weight?"  If so, please write "ppm by weight"
  17. Line 308 - "hydrogen uptake is dependent on the shape of carbon black" makes no sense. Carbon nanotubes have a completely different molecular structure, they are not carbon black in different shape. 
  18. Line 340 - delete ", as shown in Figure 6 (a)" because now you have different shape and size samples requiring 72h or about 260,000 seconds to equilibrate, unlike in that figure. 
  19. Line 352 - Table 6 <-- Table 5
  20. Line 375 - add "at 1 h after decompression" to figure caption

Response 1: After reviewing the original manuscript, we conducted a thorough check for grammar and typo errors. Subsequently, we made corrections to rectify any identified issues.

Original manuscript

Revised

Changes

This study investigated the synergistic effect of carbon black/multi-wall carbon nanotube (CB/MWCNT) hybrid fillers on the physical and mechanical properties of EPDM composites after exposure to high-pressure hydrogen gas.

This study investigated the synergistic effect of carbon black/multi-wall carbon nanotube (CB/MWCNT) hybrid fillers on the physical and mechanical properties of Ethylene propylene diene rubber (EPDM) composites after exposure to high-pressure hydrogen gas.

Line No.

15

15-16

Original manuscript

Revised

Changes

because CB causes agglomeration and has poor dispersion.

because CB particles agglomerate and have poor dispersion.

Line No.

43

44

Original manuscript

Revised

Changes

The hybrid system is to combine two or more different types of fillers in rubber composites.

The hybrid system combines two or more different types of fillers in rubber composites.

Line No.

47

48

Original manuscript

Revised

Changes

and electrical conductivity decreased compared when utilizing CB alone.

They observed that the hardness, modulus, thermal stability, and electrical conductivity increased compared when utilizing CB alone.

Line No.

68

68

Original manuscript

Revised

Changes

the volume ratio of total fillers was kept constant,

the volume fraction of total fillers was kept constant,

(I have also appropriately modified the words within the paper.)

Line No.

78

98

Original manuscript

Revised

Changes

-

We include the motivation for high-pressure hydrogen testing in the introduction of the manuscript.

Line No.

83

84-95

Original manuscript

Revised

Changes

The weights of the samples were calculated by using an electronic balance with an accuracy of 0.001 g.

The weights of the samples were determined by using an electronic balance with an accuracy of 0.001 g

Line No.

129

157

Original manuscript

Revised

Changes

χ is the polymer–solvent interaction parameter.

χ is the polymer–solvent interaction parameter (χ = 0.501).

Line No.

137

165

Original manuscript

Revised

Changes

Line No.

229

263

Original manuscript

Revised

Changes

then measured using the Flory–Rehner equation based on the swelling experiments

then estimated using the Flory–Rehner equation based on the swelling experiments.

Line No.

238

272

Original manuscript

Revised

Changes

All figures of the numbers and axis labels are mostly too tiny.

All figures of the numbers and axis labels have been enlarged.

Line No.

-

All figures (Figure 1 to Figure 9)

Original manuscript

Revised

Changes

while those of composites with MWCNT volume ratios in fillers from 4.3 to 21.5% are 417.5, 516.8, and 639.3 kPa, respectively.

while those of composites with MWCNT volume fractions in fillers from 4.3 to 21.5% are 418, 517, and 639 kPa, respectively.

Line No.

266

305

Original manuscript

Revised

Changes

the tensile properties of the hybrid composites are summarized in Table 4

the tensile properties of the hybrid composites are summarized in Table 3.

Line No.

276

327

Original manuscript

Revised

Changes

Line No.

294

347

Original manuscript

Revised

Changes

the hydrogen uptake of CNTs was 218.2 wt·ppm, whereas that of CB was higher at 326.5 wt·ppm.

the hydrogen uptake of CNTs was 218.2 , whereas that of CB was higher at 326.5 .

(is the mass of hydrogen by the mass of the samples.)

Line No.

307

360

Original manuscript

Revised

Changes

These results indicated that hydrogen uptake is dependent on the shape of carbon black.

These results indicated that hydrogen uptake is dependent on the molecular structure and shape of fillers.

Line No.

308

361

Original manuscript

Revised

Changes

We found that 72 h after hydrogen decompression was the time at which the equilibrium value of the hydrogen emission concentration was reached, as shown in Figure 6 (a).

We found that 72 h after hydrogen decompression was the time at which the equilibrium value of the hydrogen emission concentration was reached.

Line No.

339-340

393-394

Original manuscript

Revised

Changes

The changes in the mechanical properties of the EPDM/CB/MWCNT composites are shown in Figure 7 and Table 6.

The changes in the mechanical properties of the EPDM/CB/MWCNT composites are shown in Figure 8 and Table 5.

Line No.

351-352

406

Original manuscript

Revised

Changes

Figure 8. Correlation between (a) the change in tensile strength, (b) the change in elongation at break, and (c) the change in 100% modulus and hydrogen uptake.

Figure 9. Correlation between the change in the mechanical properties at 1 h after decompression and hydrogen uptake: (a) the change in tensile strength; (b) the change in elongation at break; (c) the change in 100% modulus

Line No.

375

428

7)  We calculated the units in the formulation shown in Table 1 in phr (parts per hundred of rubber). We conducted experiments by varying the MWCNT content at 0, 1, 3, and 5 phr. To observe the effects of MWCNTs in the filler hybrid system of the rubber composites, the volume fractions of MWCNT in total filler are varied from 0, 4.3, 12.9, and 21.5 % while keeping constant the volume of the total amount of fillers in rubber for all composites at 15.8 vol%.

Round 2

Reviewer 1 Report

Comments and Suggestions for Authors

NIL

Comments on the Quality of English Language

A thorough check of the grammar is advised

Reviewer 2 Report

Comments and Suggestions for Authors

The authors have the required corrections. This paper can be published as it is now.